# Electronegative LDL Promotes Inflammation and Triglyceride Accumulation in Macrophages

**DOI:** 10.3390/cells9030583

**Published:** 2020-03-01

**Authors:** Núria Puig, Lara Montolio, Pol Camps-Renom, Laia Navarra, Francesc Jiménez-Altayó, Elena Jiménez-Xarrié, Jose Luis Sánchez-Quesada, Sonia Benitez

**Affiliations:** 1Cardiovascular Biochemistry, Biomedical Research Institute Sant Pau (IIB-Sant Pau), 08041 Barcelona, Spain; npuigg@santpau.cat (N.P.); laramonun@gmail.com (L.M.); lnavarratorrecillas@gmail.com (L.N.); 2Department of Biochemistry and Molecular Biology, Faculty of Medicine, Building M, Universitat Autònoma de Barcelona (UAB), 08193 Cerdanyola del Vallès, Barcelona, Spain; 3Stroke Unit, Department of Neurology, Hospital de la Santa Creu i Sant Pau, and IIB-Sant Pau, 08041 Barcelona, Spain; pcamps@santpau.cat; 4Departament of Pharmacology. Neuroscience Institute. Faculty of Medicine, UAB, 08193 Cerdanyola del Vallès, Barcelona, Spain; francesc.jimenez@uab.cat; 5CIBER of Diabetes and Metabolic Diseases (CIBERDEM), 28029 Madrid, Spain

**Keywords:** electronegative LDL, macrophages, inflammation, foam cells, lipid droplets, TLR4, HDL, triglycerides, scavenger receptors

## Abstract

Electronegative low-density lipoprotein (LDL) (LDL(−)), a modified LDL that is present in blood and exerts atherogenic effects on endothelial cells and monocytes. This study aimed to determine the action of LDL(−) on monocytes differentiated into macrophages. LDL(−) and in vitro-modified LDLs (oxidized, aggregated, and acetylated) were added to macrophages derived from THP1 monocytes over-expressing CD14 (THP1-CD14). Then, cytokine release, cell differentiation, lipid accumulation, and gene expression were measured by ELISA, flow cytometry, thin-layer chromatography, and real-time PCR, respectively. LDL(−) induced more cytokine release in THP1-CD14 macrophages than other modified LDLs. LDL(−) also promoted morphological changes ascribed to differentiated macrophages. The addition of high-density lipoprotein (HDL) and anti-TLR4 counteracted these effects. LDL(−) was highly internalized by macrophages, and it was the major inductor of intracellular lipid accumulation in triglyceride-enriched lipid droplets. In contrast to inflammation, the addition of anti-TLR4 had no effect on lipid accumulation, thus suggesting an uptake pathway alternative to TLR4. In this regard, LDL(−) upregulated the expression of the scavenger receptors CD36 and LOX-1, as well as several genes involved in triglyceride (TG) accumulation. The importance and novelty of the current study is that LDL(−), a physiologically modified LDL, exerted atherogenic effects in macrophages by promoting differentiation, inflammation, and triglyceride-enriched lipid droplets formation in THP1-CD14 macrophages, probably through different receptors.

## 1. Introduction

Monocytes and macrophages are essential cells in host defense, as they are part of the mononuclear phagocyte system, and they are involved in both innate and adaptive immune responses. Under pathological conditions, macrophage activation leads to the release of inflammatory cytokines in response to several activators, including not only external stimuli but also endogenous signals with damage-associated molecular patterns. Immune response and inflammation are features of all stages of the atherosclerotic process, ranging from the initiation of fatty streaks to the formation of unstable plaques. During the early stages of atherosclerosis, a recruitment of monocytes towards inflammation areas occurs. Once in the vessel wall, monocytes differentiate into macrophages, which actively provide a pro-inflammatory microenvironment and contribute to remove extracellular lipids. In advanced lesions, a significant amount of neutral lipids accumulates in the cytoplasm of macrophages in specialized organelles called lipid droplets (LDs). Accumulation of LDs gives to macrophages a foamy appearance as they become lipid-laden foam cells, a hallmark of atherosclerosis and an essential element in plaque development [1,2]. The primary source of intracellular lipids is negatively-charged modified low-density lipoprotein (LDL), which macrophages uptake through scavenger receptors (SRs) [3,4]. Modified LDLs are also inductors of proinflammatory macrophage polarization, and they are abundant in atherosclerotic lesion areas [1,5]. Although the presence of modified LDLs in the circulation is minor, a naturally-occurring modified LDL, called electronegative LDL (LDL(−)), can be isolated from plasma. LDL(−) is a compendium of LDL modified by different mechanisms. Compared to native LDL, LDL(−) does not show a different level of oxidation but differs from it in many other properties [6]. LDL(−) shows differences in size and density, in lipid and protein compositions and in apoB-100 conformation (apoB). LDL(−) also has higher phospholipolytic activities, higher aggregation levels, and higher proteoglycan binding than native LDL.

LDL(−) has been proposed as a putative novel biomarker of cardiovascular risk for several reasons. LDL(−) concentration in plasma is increased in pathologies associated with cardiovascular disease, such as diabetes mellitus or familial hypercholesterolemia [6], the acute phase of myocardial infarction [7], and ischemic stroke [8]. In addition, high levels of LDL(−) have been correlated to the severity of coronary artery disease [9]. Estruch et al. reviewed on the impact that the atherogenic properties of LDL(−) have on cultured cells, particularly endothelial cells and monocytes [6]. In this regard, the main effects promoted by LDL(−) are apoptosis and inflammatory cytokine release. Although the mechanisms behind the LDL(−)-induced cell responses are not fully elucidated, some cell pathways have been described. In endothelial cells, lectin-like oxidized LDL receptor 1 (LOX-1) is essential in mediating these effects [10,11]. By contrast, the complex CD14 Toll-like receptor 4 (TLR4) is primarily responsible for triggering the intracellular pathways leading to inflammation in monocytes [6]. In this context, a cell line of monocytes overexpressing CD14 (THP1-CD14) is much more responsive to LDL(−) than the normal THP1 cell line and better reproduces the effect found in primary monocytes, in terms of the induction of inflammatory cytokines and matrix metalloproteinases (MMPs) [12,13].

Some of the properties of LDL(−) suggest that it could readily be trapped in the arterial wall, thereby allowing its interaction with macrophages in the subendothelial space. LDL(−) is prone to suffer modifications leading to aggregation [14], and it has high affinity for proteoglycans (PG) [15]. Consequently, both properties would hinder the return of LDL(−) to the bloodstream. Previous studies have reported on the atherogenic effects of LDL(−) on macrophages, mainly in relation to inflammation. Pedrosa et al. reported that LDL(−) induces apoptosis and CD36 expression in a macrophage cell line [16]. Other studies have shown that L5, the most electronegative fraction of LDL(−), releases inflammatory cytokines through LOX-1 activation in THP1-derived macrophages. In this regard, L5 isolated from infarcted patients induces interleukin 1β (IL1β) [17], granulocyte colony-stimulating factor (G-CSF), and granulocyte-macrophage colony-stimulating factor (GM-CSF) [18], and when isolated from rabbits fed an atherogenic diet, it promotes IL1β, IL6, and tumor necrosis factor α release [19]. Recently, a conformational epitope of LDL(−) has been found to activate macrophages by triggering an inflammatory response [20]. In contrast to the inflammatory action of LDL(−) on macrophages, there is lack of current studies about its potential to promote intracellular lipid accumulation. Past studies found contradictory results; some described that LDL(−) did not induce foam cell formation and was not recognized by SRs [21,22,23], whereas one study reported that LDL(−) induced foam cell formation [24].

In the current study, we aimed to investigate the effects of LDL(−) on macrophages derived from THP1-CD14 monocytes, which have a similar response to peripheral blood human monocytes in response to LDL(−) [12,25]. The main objectives were to assess the induction of an inflammatory profile and the accumulation of intracellular lipids. We also sought to make a mechanistic approach to the putative LDL(−)-induced effects on macrophages.

## 2. Materials and Methods

### 2.1. Lipoprotein Isolation

Plasma samples from healthy normolipemic subjects (total cholesterol < 5.2 mmol/L and triglyceride < 1 mmol/L) were obtained in EDTA-containing vacutainer tubes. All the subjects gave their written informed consent, and the study was conducted after approval from the Institutional Ethics Committee of the Hospital Sant Pau (IIBSP-LRB-2017-54, June 26th 2017). All lipoprotein preparations were conducted in conditions that prevented oxidation and endotoxin contamination. LDL (1.019–1.050 g/mL) and high-density lipoprotein (HDL) (1.063–1.210 g/mL) were isolated from pooled plasma by sequential flotation ultracentrifugation at 4 ºC in the presence of 1 mM EDTA and 2 µM butylated hydroxytoluene (BHT). Total LDL was separated in native LDL (LDL(+)) and LDL(−) by anion-exchange chromatography in an AKTA-FPLC system (GE Healthcare, Chicago, IL, USA). LDL(−) was concentrated using Amicon centrifugal filters (Merck Millipore, Burlington, MA, USA). In vitro-modified LDLs were used to compare their effects with that of LDL(−). Aggregated LDL (aggLDL) was obtained by mechanical shaking in a vortex for 60 s; in some of the experiments, increasing times of shaking (range: 15–90 s) were assayed. The level of aggregation was evaluated by absorbance measurement at 450 nm. Acetylated LDL (acLDL) was prepared by sequential additions of acetic anhydride, and oxidized LDL (oxLDL) was prepared by incubation with 10 µM CuSO4, as previously described [26]. 

### 2.2. Incubation of LDL with Monocytes/Macrophages

THP1-XBlue™-MD2-CD14 cells (Invivogen, San Diego, CA, USA) (THP1-CD14) are derived from a human monocytic cell line that expresses TLRs and overexpresses MD2 and CD14 receptors to increase their response to CD14-TLR ligands; in our case, LDL(−) [12]. This cell line was grown following the manufacturer’s recommendations. Flow cytometry was used to investigate the overexpression of CD14 in the THP1-CD14 monocytes in comparison to normal THP1 monocytes. The growth medium for THP1-CD14 monocytes was RPMI 1640 supplemented with 10% fetal bovine serum (FBS) and 1% Pen-Str (Biowest, Nuaille, France) and, also, with the antibiotics NormocinTM (50 mg/mL), ZeocinTM (100 mg/mL), and G418 (100 mg/mL) (Invivogen). THP1-CD14 monocytes were seeded (400,000 cells/well) with growth medium supplemented with phorbol 12-myristate 13-acetate (PMA) (Sigma, Sant Louis, MO, USA) at 50 ng/mL for 24 h to activate their differentiation into macrophages. THP1-CD14 monocytes stimulated with PMA acquired adhesive properties to plastic, and, as checked by flow cytometry, they had an increased size and a more complex state in comparison to monocytes and a loss of approximately 70% of the CD14 marker, according to those features ascribed to macrophages [27]. 

THP1-CD14 macrophages were incubated with LDL(+), LDL(−), or in vitro-modified LDLs at 60 mg apoB/L (except oxLDL at 0.2 mM total cholesterol) in RPMI 1640 supplemented with 1% FBS and 1% Pen-Str. Macrophages cultured in the same conditions but without adding LDLs were used as control cells (blank). In some experiments, cells were incubated for 1 h with a blocking antibody against TLR4 (antiTLR4) (Hycult Biotech, Frontstraat, Holanda) at 5 mg/L prior to the addition of LDLs or with HDL (60 mg apoAI/L) at the same time as LDLs. After incubation, the cell supernatants were collected and stored at −80 °C until analysis, and cells were treated as explained in the corresponding sections. Morphological appearance of macrophages was observed by bright-field microscopy at ×20 using a Zeiss Axiovert 200M inverted microscope. Images were processed by Fiji-ImageJ. The experimental procedures are summarized in Appendix A. 

### 2.3. Flow Cytometry

After incubation with LDL(+) and LDL(−), THP1-CD14 macrophages were scraped and collected in phosphate-buffered saline (PBS) containing 0.5 % bovine serum albumin (BSA). Then, flow cytometry was used to evaluate internal complexity and size (side scatter/forward scatter gating) and CD14 surface expression from this cell suspension. CD14 was quantified by incubation with 5 µL of AlexaFluor488-antiCD14 (BD Pharmingen, Franklin Laxes, NJ, USA) for 15 min. Then, 2 mL of PBS with BSA were added, and, after centrifugation, cells were resuspended in the serum-free medium and analyzed by flow cytometry. All these assays were performed in a MacsQuant Analyzer (Milteny Biotech GmbH, Bergisch Gladbach, Germany) and analyzed using MacsQuant software. 

### 2.4. Lactate Dehydrogenase (LDH) Test and ELISA

Cell viability and cytokine release were evaluated in the supernatants. Viability and proliferation were assessed by using the Cytotoxicity Detection Plus kit (Roche, Basel, Switzerland) based on the measurement of lactate dehydrogenase (LDH) activity. LDH test was conducted following the manufacturer’s recommendations. Viability was evaluated as LDH released from the cytosol of damaged cells into the supernatant. To analyze proliferation, cells were incubated with lysis buffer prior to collecting the supernatant.

The release of interleukins IL1β, IL6, IL10, and GM-CSF was quantified from supernatants by ELISA kits (Ebiosciences Santa Clara, CA, USA, for all, except Diaclone; Bensacon Cedex, France, for IL1β), as previously described [28]. 

### 2.5. Uptake of DiI-Labeled LDL(−)

LDL subfractions were labeled with the fluorescent probe 1,1′-dioctadecyl-3,3,3′,3′-tetramethylindocarbocyanine perchlorate (DiI; Molecular Probes, Eugene, OR, USA). Binding experiments were performed essentially as previously reported [13]. Cells (400,000 cells/mL) were incubated with DiI-LDLs (60 mg apoB/L) in serum-free media for 24 h at 37 °C in the absence or presence of anti-TLR4 (5 mg/L). After incubation, cells were washed with PBS, and lipid content was extracted using isopropanol. Fluorescence was measured at an excitation wavelength of 528 nm and an emission wavelength of 578 nm. 

### 2.6. Intracellular Lipid Accumulation

Intracellular lipid accumulation after incubation with stimuli was evaluated using two techniques: LD-staining with Oil Red O (ORO) and separation of lipid species by thin-layer chromatography (TLC). 

For cell staining with ORO, macrophages were fixed in the 6-well cultured plate with 4% formaldehyde for 15 min, washed with 60% isopropanol, and stained with ORO solution (0.25% in 60% isopropanol) for 30 min. Afterwards, cells were washed and then stained with hematoxylin for 30 s. The images were captured using optical microscopy at ×40 in a Zeiss Axiovert 200M inverted microscope. Alternatively, after ORO-staining, an extraction with isopropanol was performed, and absorbance was measured at 492 nm.

Lipid extraction from macrophages was performed as described [26]. The major lipid species were separated by TLC performed on silica gel plates (Macherey-Nagel, Düren, Germany) from cell lipid extracts dissolved in chloroform. TLC plates were developed using two sequential phases: heptane/diethyl ether/acetic acid (*v*/*v*/*v* 75:21:4) to 12 cm and heptane to 16 cm. The lipids were stained by dipping the plates in phosphomolybdate solution, as described in [26]. Lipids were quantified by densitometry using Quantity One 1-D Analysis software (BioRad, Hercules, CA, USA). Cell protein concentration was measured (PierceTM Rapid Gold BCA Protein Assay Kit; Thermo Fisher Scientific, Walthan, MA, USA) to normalize the intracellular lipid content.

### 2.7. Real-Time Polymerase Chain Reaction (RT-PCR)

RNA was extracted from 1 × 10^6^ cells cultured in 6-well plates. After incubation with LDLs for 4 and 24 h, cells were scraped and collected under RNase free conditions, and pellets were frozen after centrifugation. RNA extraction from cell pellets was performed with the EZ-10 DNAaway RNA miniprep kit (BioBasic; Markham, Canada). Reverse transcription was performed with 0.5 µg of RNA using EasyScript First-Stand cDNA Synthesis SuperMix (Transgen Biotech, Beijing, China). Quantitative RNA analysis was conducted using RT-PCR in a CFX96 system (BioRad, Hercules, CA, USA). All human gene expression assays were from Applied Biosystems (Beverly, MA, USA): TLR4 (Hs00152939_m1), CD36 (Hs00354519_m1), LOX-1 (Hs01552593_m1), perilipin 2 (PLIN2) (Hs00605340_m1), fatty-acid binding protein 4 (FABP4) (Hs01086177_m1), diacylglycerol acyltransferase 2 (DGAT2) (Hs01045913_m1), and carnitine palmitoyltransferase 1 a (CPT1a) (Hs00912671_m1). Human glyceraldehyde 3-phosphate dehydrogenase (GADPH) was used as the internal control (Hs99999905_m1). 

### 2.8. Statistical Analysis

Statistical analysis was performed using GraphPad Prism 6.0 software. The differences between LDL(−) and LDL(+) were tested using the Wilcoxon signed-rank test for paired data. Differences among the rest of LDL preparations were tested using the Mann-Whitney nonparametric test. Results were expressed as mean ± standard deviation (SD). A value of *p* < 0.05 was considered to be statistically significant.

## 3. Results

### 3.1. LDL(−) Induces Cytokine Release in Macrophages

LDL(−) induced cytokine release in a time and concentration-dependent manner in THP1-CD14 macrophages. In Figure 1, IL6 is shown as representative interleukin, because IL1β and IL10 had the same behavior, whereas GM-CSF followed a slightly different pattern. Figure 1a shows that there was a significant difference between LDL(−) and LDL(+) at 24 h of incubation. At 48 h, this difference remained similar for the interleukins, but it increased for GM-CSF. At 72 h, LDL(−)-induced interleukin release reached a plateau, while that of GM-CSF decreased. Figure 1b shows that at 24 h of incubation, cytokine release increased with a higher LDL(−) concentration, although IL6 release was maximum at 90 mg apoB/L, and GM-CSF release increased at higher doses. Overall, the difference between LDL(−) and LDL(+) was already observed at 60 mg apoB/L at 24 h of incubation. Therefore, these were the conditions used in the following experiments, including the incubation with in vitro-modified LDLs. 

Figure 2a shows that, in all cases, the cytokine release promoted by LDLs was higher than the macrophage basal release (blank). LDL(−) promoted a statistically significant greater cytokine release than LDL(+) and, overall, greater than in vitro-modified LDLs, being the stimulus that induced the highest IL6, IL10, and GM-CSF release. OxLDL induced higher interleukin release than LDL(+), aggLDL, and acLDL, but statistically significant differences were only achieved in comparison to LDL(+) in IL6. AggLDL and acLDL did not induce a greater effect in any of the cytokines in comparison to LDL(+). 

We aimed to determine if the high cytokine release promoted by LDL(−) was due to induction of cell mortality or proliferation. Figure 2b shows cell viability in the presence of the most inflammatory stimuli: LDL(−) and oxLDL. LDL(−) did not promote higher mortality than LDL(+) and blank. However, oxLDL decreased cell viability, an effect that was enhanced at increasing times of incubation. AcLDL and aggLDL induced a similar or slightly lower mortality rate compared to oxLDL (data not shown).

Regarding cell proliferation, no differences were found between LDL(+) and LDL(−) (88.77% ± 16.94% and 84.63% ± 17.56% at 24 h, respectively, vs. blank). This observation was corroborated by measuring the cell count and cell protein concentration levels (data not shown).

### 3.2. LDL(−) Induces Differentiation in Macrophages

The induction of GM-CSF suggests that LDL(−) could contribute to macrophage activation and differentiation. Figure 3a shows the morphological changes promoted by LDL(+) and LDL(−) on macrophages. Although both LDLs induced cell differentiation compared to nonstimulated cells, LDL(−) promoted the most differentiated appearance and presence of pseudopodia. Quantification is shown in Figure 4a. In addition, macrophages incubated with LDLs also showed a decrease in CD14 expression on their surface (Figure 3b), which is a typical feature of cell differentiation. These observations are supported by the evaluation of cell size and complexity by flow cytometry. Appendix A shows that THP1-CD14 macrophages present higher complexity compared to monocytes, particularly when incubated with LDL(−). Taken together, data suggest that LDL(−) contributes to differentiation/activation of THP1-CD14 macrophages. 

### 3.3. Role of TLR4 and HDL on LDL(−)-Induced Inflammation and Differentiation

Previously, it has been described that both TLR4 blocking and the addition of HDL attenuated the inflammatory effects of LDL(−) on monocytes [12,29]. We checked whether this behavior was also present in macrophages. Figure 4a shows that the addition of an specific antibody anti-TLR4 counteracts the differentiated state promoted by LDL(−) in macrophages by diminishing the number of pseudopodia, whereas it was not different in the case of LDL(+)-induced cell differentiation. In addition, Figure 4b shows that blocking of TLR4 also decreased LDL(−)-induced cytokine release, which was statistically significant for IL6 and GM-CSF. The addition of anti-TLR4 had no effect on cell viability, as checked by a LDH test (differences lower than 2% vs. blank). 

The addition of HDL inhibited the release of IL1β, IL6, and GM-CSF promoted by LDL(−) at 24 h in human macrophages (Figure 4c). By contrast, HDL had no inhibitory effect on LDL(−)-induced IL10 release. The addition of HDL did not modify cell differentiation or viability (differences in viability lower than 3% vs. blank).

### 3.4. LDL(−) is More Avidly Internalized than LDL(+) and it Induces LD Formation

The uptake of LDL(−) was two-fold higher than that of LDL(+), and it was not inhibited by anti-TLR4 (Figure 5a), which discards the involvement of TLR4 in LDL internalization. On the other hand, LDL(−) uptake was displaced by increasing concentrations of oxLDL, whereas LDL(+) was not (Figure 5b), suggesting a role for scavenger receptors in LDL(−) uptake. 

To ascertain the role of LDL(−) internalization in the conversion of macrophages into foam cells, LDL(−) was added at increasing incubation times (24 h, 48 h, and 72 h), and its effect was compared with that promoted by in vitro-modified LDLs. After incubation with the stimuli, LDs were stained with ORO. Figure 6 shows images of macrophages incubated in the presence or absence of LDLs for 24 and 48 h. A higher reddish staining was seen in the presence of the LDLs than in the basal condition; however, LDL(+) induced the lowest amount of staining in comparison to the other LDLs. The ORO-positive LDs promoted by LDLs were more abundant at 48 h than at 24 h of incubation. Among in vitro-modified LDLs, aggLDL induced the highest LD-staining in comparison to oxLDL and acLDL (both shown in Appendix A), which are also known to induce foam cell formation. However, LDL(−) was the stimulus that induced the highest lipid accumulation, even in comparison to aggLDL. The amount of lipid accumulation was quantified by the absorbance measurement at 492 nm after organic extraction. In comparison to LDL(+), this quantification corroborated that LDL(−) had the greatest impact on inducing LDs, as indicated in Figure 6. 

The higher aggregation level of LDL(−) compared to LDL(+) [14] could contribute to LDL(−)-induced lipid accumulation. In the current study, we found that the aggregation level of LDL(−) and aggLDL (after 60 s of vortexing) was 1.5-fold and 5-fold, respectively, compared to that of LDL(+)., as shown in Appendix A.; however, LDL(−) was more a powerful inductor of LD generation than aggLDL. We performed additional experiments after inducing LDL(+) and LDL(−) in vitro aggregation via vortexing for increasing times (15 s, 30 s, 60 s, and 90 s). Appendix A shows LD-staining at 15 s and 60 s of aggregation of both LDLs (representative of low and high LDL aggregation). LDL(+) that was vortexed for less than 60 s had low ability to induce LDs. The maximum effect was reached at 60 s, and it remained similar at 90 s (not shown). LDL(−) in basal state induced more LD formation than LDL(+) aggregated at any of the assayed times, and this effect was even higher when the aggregation was induced, as shown at 15 s and 60 s in Appendix A. 

Appendix A shows that, in contrast to its anti-inflammatory effect, HDL did not revert the induction of intracellular lipid accumulation promoted by LDL(−).

### 3.5. LDL(−) Induces Particularly Triglyceride (TG) Accumulation

TLC was used to ascertain the specific lipids whose intracellular accumulation was induced by LDL(−). Figure 7 shows the separation of the major lipid species at 24 h, 48 h, and 72 h of incubation with LDL(−), aggLDL, and LDL(+). AggLDL was the in vitro-modified LDL that induced the highest lipid accumulation. LDL(−) induced higher esterified cholesterol (EC) accumulation than LDL(+)-incubated macrophages,. Additionally, the most noteworthy observation was that LDL(−) strongly induced the accumulation of triglyceride (TG), an effect that was promoted in a time-dependent manner. The next experiments were conducted at 24 h and 48 h, since 72 h was discarded because of the decrease in cell viability. 

The quantification of intracellular EC and TG found by TLC at 24 h and 48 h is shown in Figure 8a,b, respectively. Figure 8a shows that both aggLDL and LDL(−) induced intracellular EC accumulation compared to blank and LDL(+)-stimulated macrophages. No statistical differences were observed in free cholesterol (FC) cell content (data not shown). The remarkable effect of LDL(−) in TG accumulation was confirmed when the intensity of the TLC spots was quantified, as shown in Figure 8b. LDL(−) promoted the greatest TG accumulation in macrophages compared to the other stimuli, its effect significantly higher than that of blank, LDL(+), and aggLDL at both 24 h and 48 h. Taken together, the relative content of intracellular TG/EC was twice as high when the cells were incubated with LDL(−) than when they were incubated with aggLDL. Blocking of TLR4 did not diminish LDL(−)-induced lipid accumulation in a statistically significant manner (Figure 8c).

### 3.6. LDL(−) Promotes Changes in Gene Expression

Gene expression assays were performed in order to make a mechanistic approach to the pathways involved in TG accumulation promoted by LDL(−). Figure 9a,b show the expression of genes related to TG metabolism and of cell receptors, respectively, at 4 h of incubation. Gene expression was lower at 24 h than at 4h for all the evaluated genes (data not shown). 

LDL(−) induced upregulation of *FABP4* and *PLIN2* expression vs. LDL(+); it also induced upregulation of *DGAT2*, although in a nonsignificant manner. 

Both lipid accumulation and inflammation induced by LDL(−) in macrophages could be promoted through the increased expression of specific cell receptors mediating these effects. LDL(−) induced *CD36* upregulation, and it had a trend to increase the expression of *LOX-1* compared to LDL(+). However, no differences were found in *TLR4* expression.

## 4. Discussion

LDL(−) is a naturally occurring modified LDL present in human plasma. Several actions of LDL(−) play a key role in the development of atherosclerosis, mainly the induction of inflammatory cytokines in endothelial cells [30] and the release of cytokines and MMPs in monocytes [13,31]. These effects trigger the recruitment of activated monocytes to the arterial wall, where they can differentiate into macrophages, which are hallmark cells in atherosclerotic lesions. In the arterial wall, macrophages could interact with LDL(−), since this lipoprotein is easily retained by PG [15]. In this regard, the importance and novelty of the current study is that macrophage activation is induced by LDL(−), a physiologically modified LDL present in blood. It promoted inflammation and differentiation of THP1-CD14 macrophages and contributed to their transformation into lipid-laden foam cells.

Our previous studies reported that, in the presence of LDL(−), THP1 monocytes overexpressing CD14 (THP1-CD14) reproduced the inflammatory response of human monocytes isolated from peripheral blood [12,31], whereas LDL(−) induced a minimal inflammatory effect in normal THP1 monocytes. This latter behavior was attributed to low CD14 and TLR4 expression, which are key receptors in the LDL(−)-induced inflammatory effect on monocytes [12]. In the present study, the effect of LDL(−) was assessed in THP1-CD14-derived macrophages. The high CD14 expression on THP1-CD14 monocytes diminished when they differentiated into macrophages, as previously described for the differentiation of human monocytes in circulation [27]. THP1-CD14 macrophages showed adhesive properties and a differentiated state. Taken together, these observations suggest that THP1-CD14 is a valid cell model, and, in this context, the behavior of THP1-CD14 macrophages resembles that of macrophages in physiological conditions. 

LDL(−) induced differentiation in THP1-CD14 macrophages by promoting morphological changes that are characteristic of activated macrophages, such as the presence of elongated pseudopodia, as well as the loss of cell surface CD14 compared to nonstimulated macrophages. The electronegative subfraction L5 has recently been reported to also induce differentiation of THP1 monocytes into macrophages [32]. This effect could be partly promoted by the ability of LDL(−) to activate p38 MAPK and CREB [28], both inductors of cell differentiation [33], and through the release of GM-CSF, which triggers the differentiation and activation of macrophages [34]. The finding of LDL(−)-induced GM-CSF release is in agreement with a study performed on THP1 macrophages with the most electronegative subfraction (L5) [18].

In our study, compared to other in vitro-modified LDLs, LDL(−) was the main inductor of GM-CSF, IL6, and IL10 release, and it had a similar effect on IL1β as oxLDL. This occurred in the absence of changes in cell proliferation or mortality. According to our results, in THP1 macrophages, L5 from infarcted patients also induced IL1β release [17], and L5 from rabbits fed an atherogenic diet induced IL1β and IL6 [19]. The induction of IL10, a cytokine that suppresses macrophage activation and inflammation [35], could regulate an excessive inflammatory response to LDL(−), as was previously described in circulating monocytes [31]. Interestingly, HDL attenuated the LDL(−)-induced inflammatory response, since it did not alter IL10 release but diminished the release of the inflammatory cytokines, as described in monocytes [15]. Blocking of TLR4 also inhibited LDL(−)-induced inflammatory cytokine release and cell differentiation. Hence, TLR4 activation by LDL(−) seems to be essential in mediating these effects in macrophages. 

Taken together, the induction of inflammation by LDL(−), both in monocytes and in macrophages, would elicit the recruitment of monocytes to the arterial wall, thus leading to their differentiation and activation. In this context, the second finding of the current study, which is the accumulation of intracellular LDs in macrophages induced by LDL(−), is especially outstanding. This action seems to be promoted because of the high internalization of LDL(−) in comparison to LDL(+). In our study, the poor displacement by oxLDL of LDL(+) uptake and the low LDL(+)-induced lipid accumulation are in agreement with the well-known behavior of native LDL, which is not recognized by SR but by LDL receptor, whose expression is regulated by intracellular cholesterol concentrations. Interestingly, lipid accumulation induced by LDL(−) followed a different pattern than that of other modified LDLs, since LDL(−) lead to the formation of LDs enriched in TG rather than in EC, in contrast to that described for aggLDL, ox LDL, and acLDL [3,36]. Moreover, TG accumulation was promoted in a time-dependent manner; this suggests that this process is not down-regulated over the assay timeframes. This surprising ability of LDL(−) to store intracellular TG has also been described in cardiomyocytes [37]. LDL(−)-induced TG-enriched LDs are probably not only due to the TG content in LDL(−), which is much lower than that of cholesterol, but also to their increased non-esterified fatty acid (NEFA) content, which can contribute to TG synthesis. 

The induction of lipid accumulation by LDL(−) could be a consequence of different mechanisms: (1) higher LDL(−) uptake by specific cell receptors, (2) activation of the pathways involved in TG and/or EC synthesis and storage, and (3) inhibition of the pathways involved in TG/EC hydrolysis and in lipid efflux. Therefore, we analyzed the expression of genes codifying for receptors of modified LDL and for proteins involved in TG metabolism. It has been reported that TLR4 activation promoted TG accumulation in macrophages through DGAT2 and other TG metabolism-related enzymes [38,39]. However, in the present study, based on the lack of statistically significant changes in the presence of anti-TLR4, TLR4 did not play an important role in the uptake of LDL(−) or in LDL(−)-induced intracellular TG. Moreover, LDL(−) did not alter *TLR4* expression. Therefore, the cytokine release could be promoted by LDL(−) activating TLR4 to a higher extent than LDL(+), as described for monocytes [12], without the uptake of LDL(−). However, LDL(−) upregulated *LOX-1* and *CD36* expression, as reported in other monocyte/macrophage cell lines [16,32]. Both receptors are related to modified LDL uptake, inflammation, and foam cell formation [40,41]. Their expression increases in differentiated macrophages vs. monocytes, as they are highly expressed in lipid-laden macrophages, and particularly in the presence of stimuli, such as OxLDL [42,43]. Previous studies have reported the involvement of these receptors in some LDL(−) actions in macrophages [16,17]. Taken together, these observations suggest the involvement of LOX-1 and CD36 in the LDL(−) effects described in the current study. However, other receptors, such as SRA and LRP1 [44], and nonreceptor-mediated processes, such as macropinocytosis [45], could also play a role. Displacement of LDL(−) uptake by oxLDL suggests that both LDLs share common receptors; however, the different pattern of lipid accumulation promoted by LDL(−) compared with that promoted by oxLDL and other modified LDLs suggests that, once internalized, LDL(−) induces lipid accumulation by specific mechanisms. The intracellular pathways leading to TG accumulation could be activated owing to the physiological properties of LDL(−). To ascertain the mechanisms of the binding and uptake of LDL(−), as well as the relationship with the activated intracellular pathways, future studies must be addressed.

Regarding the molecules implicated in lipid storage, LDL(−) induced *FABP4* and *PLIN2* upregulation. The expression of both molecules is induced during differentiation from monocytes into macrophages and by modified LDL [36,46]. FABP4 is a chaperone that transports fatty acids, and it is involved in insulin resistance, inflammation, and foam cell formation [46]. PLIN2, the most abundant LD-associated protein in macrophages, prevents lipid efflux and regulates lipid retention, including TG accumulation, eventually leading to foam cell formation [47]. Furthermore, LDL(−) showed a trend to induce the expression of *DGAT2*, an enzyme involved in the last step of TG synthesis; then, it probably also contributes to the accumulation of TG induced by LDL(−). However, LDL(−) did not alter *CPT1a*, which codifies for an enzyme that transports fatty acids to mitochondria for their β-oxidation. These observations suggest that NEFA, generated from the intracellular degradation of LDL(−), are preferably used for TG synthesis rather than for energy generation.

As discussed above, the action of LDL(−) on macrophages is probably a consequence of its physicochemical properties; its lipid composition and high aggregation level could be important contributors in this regard. The increased content of NEFA, ceramide, and sphingosine in LDL(−) has been related to the high aggregation of LDL(−) and to its inflammatory effect on monocytes [29,48,49]. In the current study, LDL(−)-induced cytokine release in macrophages was hindered in the presence of HDL, which had been previously described as exerting a protective response in monocytes by diminishing NEFA content and the aggregation level of LDL(−) [29]. Therefore, although aggregation does not seem to be the only factor contributing to the atherogenic effects of LDL(−) on macrophages, it probably plays an important role. In this regard, LDL(−) could be further aggregated in the subendothelial microenvironment, where it is highly retained [15], and it could acquire further atherogenic potential. In this line of evidence, Lehti et al. reported that LDL isolated from carotid plaque showed a high aggregation level and induced TG accumulation in macrophages [50]. 

Inflammation and foam cell formation are essential processes in atherosclerosis, and the relationship between them is complex, probably depending on the physiological context. Although some studies have described that excessive cholesterol accumulation lead to inflammation [51], other studies suggest that the inflammatory response can precede and lead to lipid accumulation [52]. In this sense, the cytokine release promoted by LDL(−) could contribute to lipid accumulation, i.e., through the production of GM-CSF, an inductor of cell differentiation [53]. However, the presence of foam cells has been associated with the suppression of inflammatory genes [54]. In a physiological context, lipid body formation induced by LDL(−) could be considered to be an initial cellular defense against lipotoxicity to remove harmful lipids, such as NEFA [55], as well as a regulatory mechanism, by which the release of inflammatory mediators is under control [56]. It is noteworthy that, in the present study, both inflammatory and foam cell formation effects were promoted by LDL(−) in the absence of mortality and that LDL(−) was isolated from healthy subjects and at a concentration similar to that found in plasma. 

## 5. Conclusions

In summary, LDL(−) can induce inflammation in differentiated macrophages and promote the accumulation of TG-enriched LDs. In comparison to in vitro-modified LDL, LDL(−) induced these atherogenic effects to a greater extent and in a different manner. The current results suggest the involvement of several cell receptors and cross-talk among several LDL(−)-activated pathways in macrophages. The complex interaction between these pathways should be addressed in future studies. Taken together, our findings highlight new significant actions of LDL(−) on macrophage activation in the context of the development of atherosclerosis.

## Figures and Tables

**Figure 1 cells-09-00583-f001:**
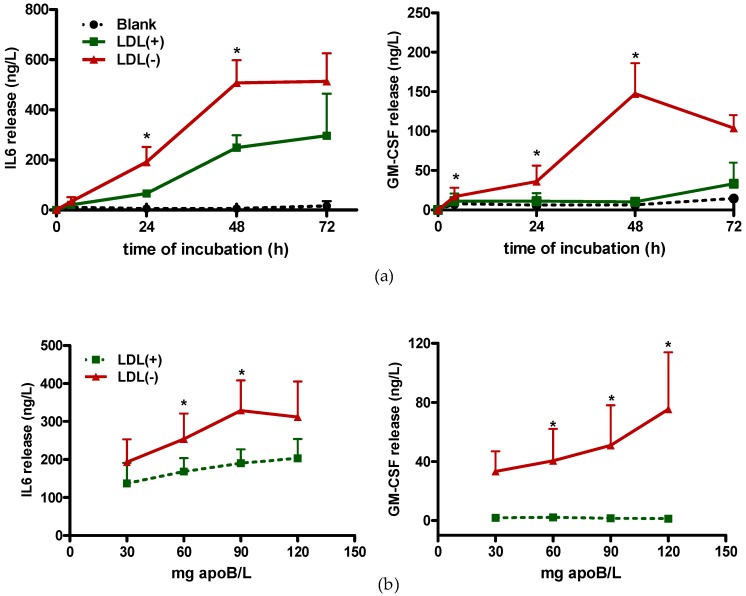
Time-course and dose-response to low-density lipoprotein (LDL)(−)-induced cytokine release. THP1-CD14 macrophages (400,000 cells/mL) were incubated with LDL(+) (green line) or LDL(−) (red line) (60 mg apoB/L), or in the absence of stimuli (blank: black dotted line), for the indicated times (**a**) and for 24 h at the indicated LDL concentrations (**b**). Cytokines were evaluated in the supernatant by ELISA. Results are expressed as ng/L, mean ± SD (*n* = 6 for time-course and *n* = 4 for dose-response), * vs. LDL(+), and *p* < 0.05.

**Figure 2 cells-09-00583-f002:**
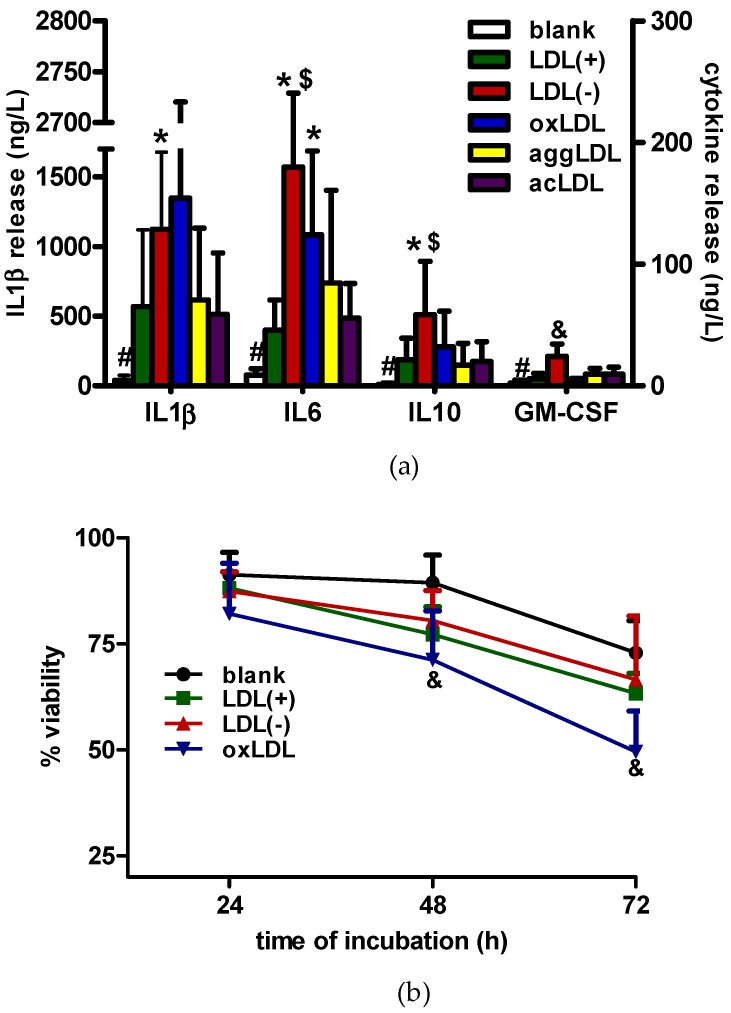
Cytokine release and cell viability in the presence of LDLs. THP1-CD14 macrophages (400,000 cells/mL) were incubated with LDLs (60 mg apoB/L) or in the absence of stimuli for 24 h. (**a**) Cytokine release evaluated in the supernatant by ELISA. Results are expressed as ng/L; mean±SD (*n* = 7); * vs. LDL(+); # blank vs. stimuli (LDLs); & vs. LDL(+), oxLDL, aggLDL and acLDL.; $ vs. aggLDL and acLDL; and *p* < 0.05. (**b**) Cell viability measured by an enzymatic lactate dehydrogenase (LDH) kit in the supernatant. Results are expressed as % cell viability, mean ± SD (*n* = 6), & vs. blank, and *p* < 0.05.

**Figure 3 cells-09-00583-f003:**
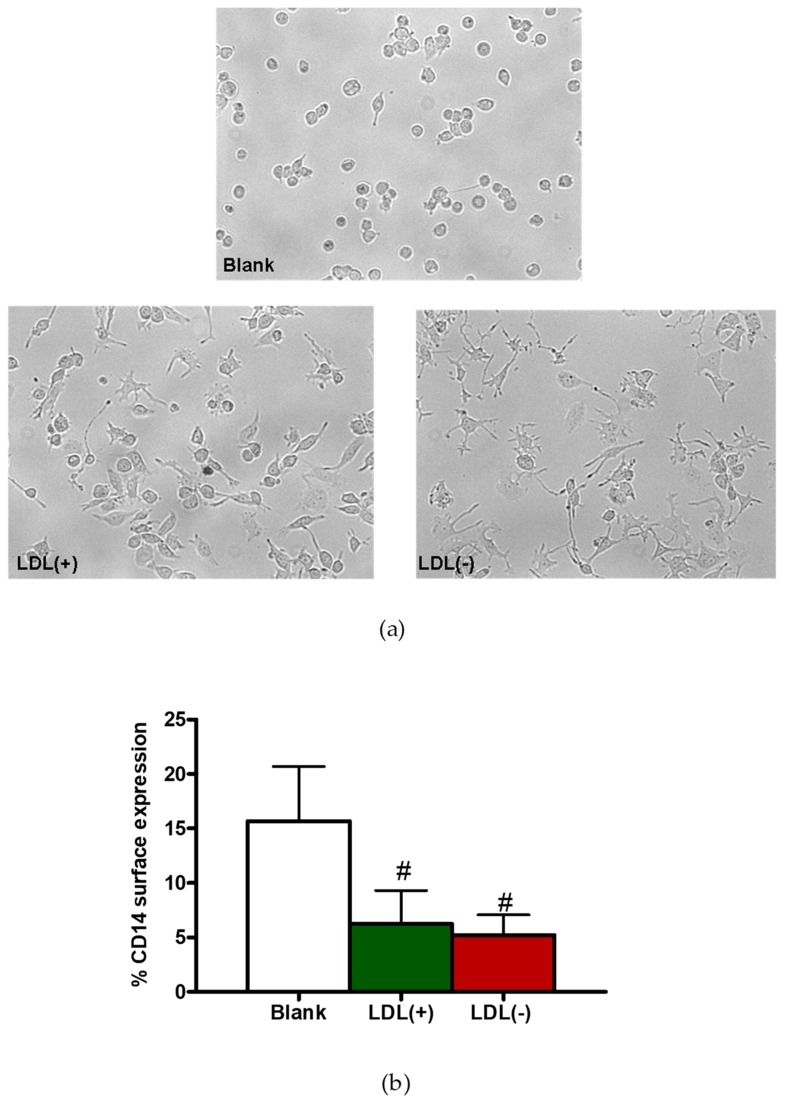
Cell differentiation in the presence of LDL(+) and LDL(−). THP1-CD14 macrophages (400,000 cells/mL) were incubated with LDL(+) or LDL(−) (60 mg apoB/L) for 24 h. Morphological changes were observed by bright-field microscopy (491.82 × 367.45 µm) (**a**) and CD14 surface expression by flow-cytometry (**b**). In (b), results are expressed as % of cells expressing CD14, mean ± SD (*n* = 4), # vs. blank, and *p* < 0.05.

**Figure 4 cells-09-00583-f004:**
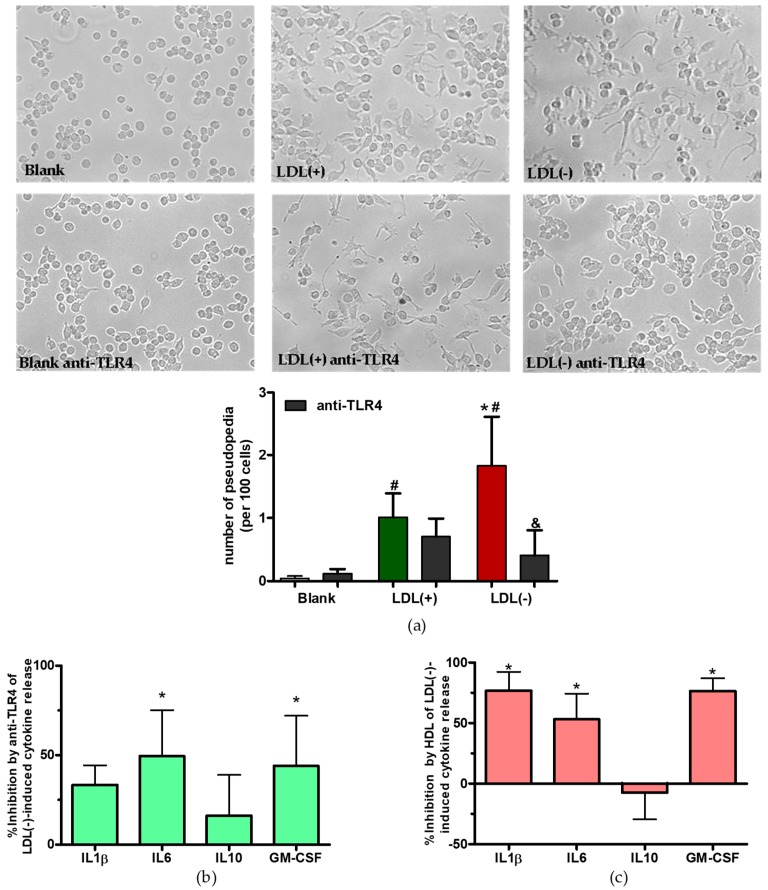
Effect of anti-TLR4 and high-density lipoprotein (HDL) in LDL(−)-induced differentiation and inflammation. THP1-CD14 macrophages (400,000 cells/mL) were incubated with LDLs (60 mg apoB/L) in the presence or absence of anti-TLR4 (5 mg/L) or HDL (60 mg apoAI/L) for 24 h. Cell morphology was observed by bright-field microscopy (491.82 × 367.45 µm), and the number of pseudopodia was quantified by Fiji-ImageJ (*n* = 4), # vs. blank, * vs. LDL(+), & vs. absence of anti-TLR4, and *p* < 0.05. (**a**). Inhibition of LDL(−)-induced cytokine release by the presence of anti-TLR4 and HDL vs. the absence of both is shown in (**b**) and (**c**), respectively. Results are expressed as %, mean ± SD (*n* = 6 for HDL and *n* = 7 for TLR4), * vs. absence of anti-TLR4 and HDL, and *p* < 0.05.

**Figure 5 cells-09-00583-f005:**
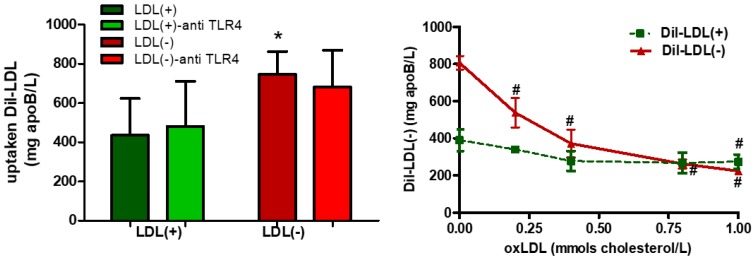
LDL(−) uptake in the presence or absence of anti-TLR4 and oxLDL. THP1-CD14 macrophages (400,000 cells/mL) were incubated with DiI-LDL(+) and DiI-LDL(−) (60 mg apoB/L) in the presence or absence of anti-TLR4 or of oxLDL for 24 h. Then, fluorescence was measured from lipid extraction of macrophages. Results are expressed as mg apoB/L of DiI-LDL, mean ± SD. (**a**) DiI-LDL(+) and DiI-LDL(−) uptake were evaluated in the presence or absence of anti-TLR4 (5 mg/L) (*n* = 6), * vs. LDL(+), and *p* < 0.05. (**b**) DiI-LDL(+) and DiI-LDL(−) uptake were evaluated in the presence or absence of increasing concentrations of oxLDL (0–1 mmols cholesterol/L (*n* = 4), # vs. absence of oxLDL, and *p* < 0.05.

**Figure 6 cells-09-00583-f006:**
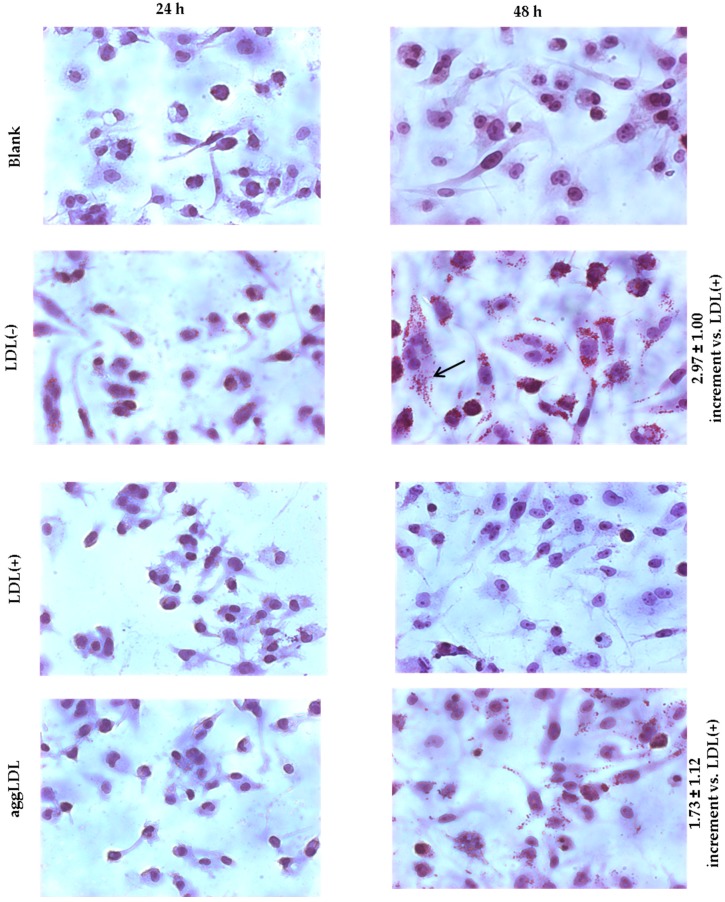
Lipid droplet (LD)-staining with oil red O (ORO). THP1-CD14 macrophages were seeded in 6-well plates (2 mL, 400,000 cells/mL) and incubated with LDLs (60 mg apoB/L) for 24 h and 48 h. Afterwards, cells were fixed and stained with ORO and were observed by light-field microscopy (249.00 × 186.04 µm). Representative image of *n* = 5. The staining at 48 h was quantified by the absorbance measurement in lipid extracts, and the increase vs. LDL(+) is indicated in the figure (right side).

**Figure 7 cells-09-00583-f007:**
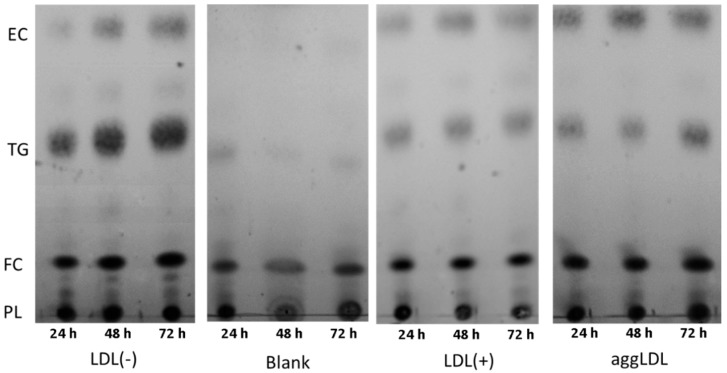
Intracellular lipid separation by thin-layer chromatography (TLC). THP1-CD14 macrophages were seeded in 6-well plates (2 mL, 400,000 cells/mL) and incubated with LDL(+), LDL(−), or aggLDL (60 mg apoB/L) for 24 h, 48 h, and 72 h. Afterwards, lipids were extracted and separated by TLC using two mobile phases. EC: esterified cholesterol, TG: triglycerides, FC: free cholesterol, and PL: phospholipids. Representative image of *n* = 3.

**Figure 8 cells-09-00583-f008:**
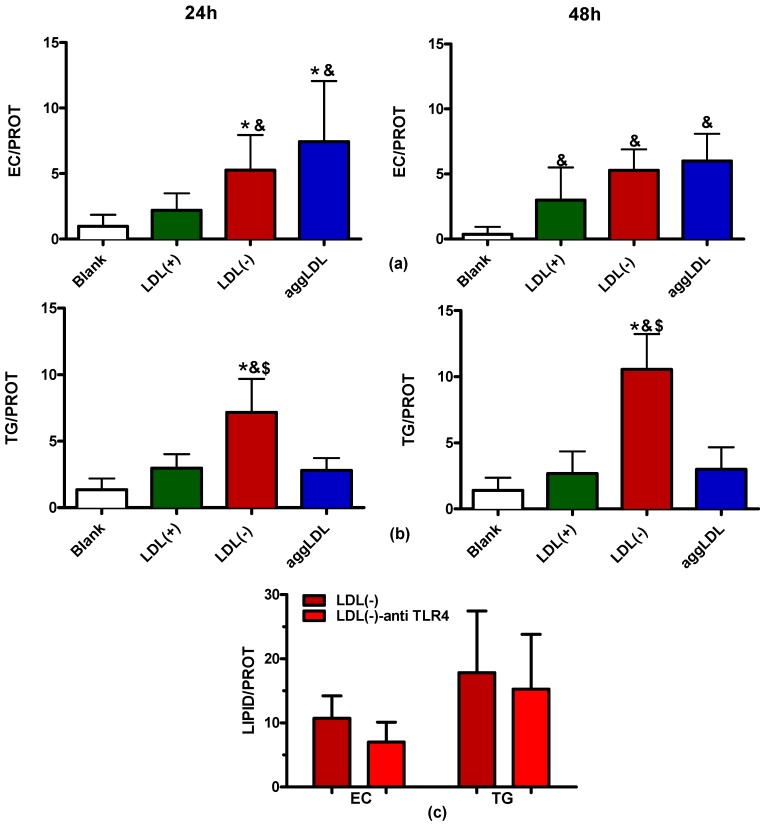
Intracellular lipid accumulation. THP1-CD14 macrophages were seeded in 6-well plates (2 mL, 400,000 cells/mL) and incubated with LDL(+) (green bars), LDL(−) (red bars), or aggLDL (blue bars) at 60 mg apoB/L for 24 h and 48 h. Afterwards, lipids were extracted and separated by TLC using two mobile phases. The TLC plates were scanned, and the intensity of the bands corresponding to EC (**a**) and TG (**b**) was quantified. (**c**) EC and TG were also quantified in macrophages incubated with LDL(−) in the presence of anti-TLR4 for 48 h. Results are expressed as EC or TG vs. cell protein content, mean ± SD (*n* = 5), * vs. LDL(+), & vs. blank, $ vs. aggLDL, and *p* < 0.05.

**Figure 9 cells-09-00583-f009:**
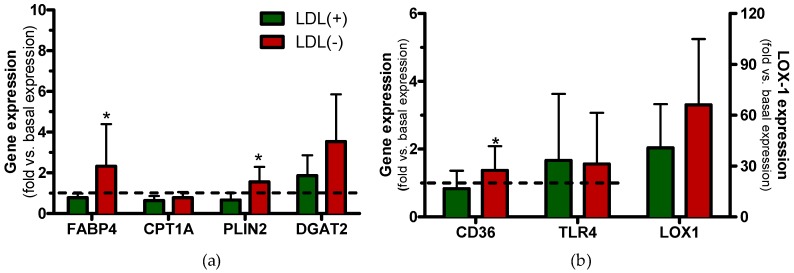
Gene expression in the presence of LDL(+) and LDL(−). THP1-CD14 macrophages were seeded in 6-well plates (1 mL, 1,000,000 cell/mL) and incubated with LDL(+) (green bars) or LDL(−) (red bars) at 60 mg apoB/L for 4 h. Afterwards, RNA was extracted, cDNA was generated, and the number of relative RNA copies was quantified by RT-PCR performed with probes of the indicated genes ((**a**): TG-related genes and (**b**): cell receptors). *GADPH* was used as the internal control (constitutive expression), and the number of relative RNA copies of the genes was expressed vs. *GADPH*. Results are expressed as fold (in the number of RNA copies) vs. basal expression (blank), mean ± SD (*n* = 7), * vs. LDL(+), and *p* < 0.05.

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
