# Peer review of "Electronegative LDL Promotes Inflammation and Triglyceride Accumulation in Macrophages"

_cells, 2020, doi:10.3390/cells9030583_

Round 1

Reviewer 1 Report

Electronegative LDL, such as LDL(-), has been proposed to play a pathogenic role in atherosclerosis from the perspectives of cell, animal, and human studies. In this well-performed and nicely written work, the authors, who are world experts in this field, demonstrate new evidence that LDL(-) induces expression and release of inflammatory cytokines and promotes triglyceride accumulation in macrophages. The most important contribution of the paper is the clear demonstration of lipid drop formation comprising triglycerides and esterified cholesterol. The concomitantly induced expression of FABP4 and PLIN2 further strengthens the observation. The findings strongly substantiate the potential role of electronegative LDL in atherosclerosis.

It would be interesting in their future studies to examine whether blocking LOX-1 can attenuate TG accumulation in macrophages on LDL(-) exposure. It would also be of interest to know if similar findings (TG accumulation) can be observed in arterial endothelial cells and if so, whether the fat accumulation is associated with endothelial cell apoptosis.

Minor error:

Figure S1. LD staining by oxLDL and acLDL. THP1-CD14 macrophages were seeded in 6-well plates (2 ml, 400,000 cells/ml) and incubated with oxLDL or acLDL(-) (60 mg apoB/L) for 48 h.  

Possible error: acLDL(-) should be acLDL.

Author Response

Reviewer 1

Electronegative LDL, such as LDL(-), has been proposed to play a pathogenic role in atherosclerosis from the perspectives of cell, animal, and human studies. In this well-performed and nicely written work, the authors, who are world experts in this field, demonstrate new evidence that LDL(-) induces expression and release of inflammatory cytokines and promotes triglyceride accumulation in macrophages. The most important contribution of the paper is the clear demonstration of lipid drop formation comprising triglycerides and esterified cholesterol. The concomitantly induced expression of FABP4 and PLIN2 further strengthens the observation. The findings strongly substantiate the potential role of electronegative LDL in atherosclerosis.

It would be interesting in their future studies to examine whether blocking LOX-1 can attenuate TG accumulation in macrophages on LDL(-) exposure. It would also be of interest to know if similar findings (TG accumulation) can be observed in arterial endothelial cells and if so, whether the fat accumulation is associated with endothelial cell apoptosis.

Thank you for the kind comments. Of course, future studies analyzing in depth the role of LOX1 in LDL(-) uptake in macrophages are needed. Regarding endothelial cells, although a priori it is not expected that these cells accumulate lipid droplets due to lack of the related intracellular pathways, the idea is interesting and deserves some studies to discard or not this possibility.

Minor error:

Figure S1. LD staining by oxLDL and acLDL. THP1-CD14 macrophages were seeded in 6-well plates (2 ml, 400,000 cells/ml) and incubated with oxLDL or acLDL(-) (60 mg apoB/L) for 48 h.  

Possible error: acLDL(-) should be acLDL.

The reviewer is right, the sample was acLDL; accordingly, the Figure Legend has been corrected.

Reviewer 2 Report

     In this study, the authors investigate the effects of electronegative LDL [LDL(-)] on monocyte/macrophages using a monocytic cell line overexpressing CD14 (THP1-CD14).  The effect of LDL(-) on the cytokine secretion , cell differentiation, lipid accumulation and gene expression were explored and compared to that of other modified forms of LDL.  The authors report that LDL(-) was more effective at inducing cytokine secretion (IL-1β, IL-6, IL-10 and GM-CSF) than other modified LDLs.  Moreover, LDL(-) induced morphological changes associated with macrophage differentiation.  These effects were antagonized by an anti-TLR4 antibody and HDL.  LDL(-) was also more effective, when compared to LDL(+), to promote intracellular lipid accumulation in into triglyceride-enriched droplets, an effect that was inhibited by oxidized LDL, but not anti-TLR4 or HDL, suggesting that lipid uptake pathways are separate from inflammatory pathways.  Finally, it is reported that LDL(-) promoted the expression of scavenger receptors (CD36 and LOX-1) and several genes involved in triglyceride accumulation.              

            This is a thorough and interesting study analyzing several effects of LDL(-) on macrophages that may have an impact in the atherosclerotic process.  The experimental design of the study is solid and the results and clearly presented.  The manuscript is very well-written.  There are only a few minor revisions recommended.

The authors write an extensive Introduction addressing the rationale of the study and the potential mechanisms whereby LDL(-) may have a role in atherosclerosis. One aspect missing, however, is description of the molecular aspects of LDL(-) and its presence in plasma.  The authors may want to include a few brief paragraphs describing LDL(-).  Spell BHT (Materials and Methods, line 102) On line 126, what do the authors mean by “deficient-FBS serum”? Is it lipid-deficient FBS?  Low concentration FBS medium?  The method describing the staining of the cells with Oil-Red O (line 167) doesn’t mention how the cells were prepared before the staining. Was it cytospin slides?  This should be included.  In Figure 1(a), the symbols corresponding to LDL(+) in the GM-CSF graph (right side) are triangles, just like LDL(-), when it should have been squares according to the symbol key. 

Author Response

 In this study, the authors investigate the effects of electronegative LDL [LDL(-)] on monocyte/macrophages using a monocytic cell line overexpressing CD14 (THP1-CD14).  The effect of LDL(-) on the cytokine secretion , cell differentiation, lipid accumulation and gene expression were explored and compared to that of other modified forms of LDL.  The authors report that LDL(-) was more effective at inducing cytokine secretion (IL-1β, IL-6, IL-10 and GM-CSF) than other modified LDLs.  Moreover, LDL(-) induced morphological changes associated with macrophage differentiation.  These effects were antagonized by an anti-TLR4 antibody and HDL.  LDL(-) was also more effective, when compared to LDL(+), to promote intracellular lipid accumulation in into triglyceride-enriched droplets, an effect that was inhibited by oxidized LDL, but not anti-TLR4 or HDL, suggesting that lipid uptake pathways are separate from inflammatory pathways.  Finally, it is reported that LDL(-) promoted the expression of scavenger receptors (CD36 and LOX-1) and several genes involved in triglyceride accumulation.              

            This is a thorough and interesting study analyzing several effects of LDL(-) on macrophages that may have an impact in the atherosclerotic process.  The experimental design of the study is solid and the results and clearly presented.  The manuscript is very well-written.  There are only a few minor revisions recommended.

The authors write an extensive Introduction addressing the rationale of the study and the potential mechanisms whereby LDL(-) may have a role in atherosclerosis. One aspect missing, however, is description of the molecular aspects of LDL(-) and its presence in plasma.  The authors may want to include a few brief paragraphs describing LDL(-). 

Thank you for your kind and constructive comments. Description and molecular properties of LDL(-) have been included in Introduction.

Spell BHT (Materials and Methods, line 102)

Spell of BHT has been added

On line 126, what do the authors mean by “deficient-FBS serum”? Is it lipid-deficient FBS?  Low concentration FBS medium? 

It means 1% FBS, the text has been modified accordingly.

The method describing the staining of the cells with Oil-Red O (line 167) doesn’t mention how the cells were prepared before the staining. Was it cytospin slides?  This should be included. 

Regarding staining of cells with ORO, cytospin slides were not used. After incubating with the stimuli, supernatant was collected and macrophages were fixed in the same cell well with 4% formaldehyde, and afterwards they were stained as described in Methods.

In Figure 1(a), the symbols corresponding to LDL(+) in the GM-CSF graph (right side) are triangles, just like LDL(-), when it should have been squares according to the symbol key.  

The mistake in Figure 1 (a) has been corrected

Reviewer 3 Report

This manuscript aims to study the pro-atherogenic effects of electrogenative LDL (LDL(-)) on the THP-1-CD14 macrophage-like cell line. The authors have presented data showing that LDL(-) induced cytokine release, neutral lipid accumulation (especially triglycerides), and some genes involved in lipid metabolisms. In addition, they also compared the effects of LDL(-) with other similar but well-known atherogenic lipoproteins such as acetylated LDL, oxidized LDL (oxLDL). Unfortunately, the whole story appears incremental, as all the effects shown here are well known induced by oxLDL for many years. The molecular mechanism underlying the effects of LDL(-) is very under-developed in this work.

Major issues:

What is the rationale for using the THP1-CD14 cell line instead of primary macrophages for the whole study? There are many well-established protocols for isolation, culturing and treating primary macrophages (either mouse or human species). Since the THP1 cell line cannot fully mimick the behavior of macrophages, data from this cell line should be interpreted with caution. Effects of minimally modified LDL (mmLDL) including oxLDL, Acetylated LDL on macrophages are known for more than decades. These effects include cytokine release, intracellular lipid accumulation leading to foam cell formation, alteration in scavenger receptor expression (e.g. CD36, LOX-1, etc) and lipid metabolism-related gene expression. The LDL(-), based on the author’s description, seems to be one form of mmLDL. So I really do not see much novelty in this study. Fig. 3 and Fig. 4 on cell differentiation are not convincing. Specific differentiation markers should be used and quantified by flow cytometry or immunoblots instead of just bright-field images. Additionally, since the anti-TLR4 antibody data are all negative, it appears that either the TLR4/CD14 are dispensable for the effects here or the anti-TLR4 antibody is not working. But no data is showing to validate whether anti-TLR4 antibody is good or not, no any conclusion can be drawn here.

Reviewer 4 Report

In the present work the authors have demonstrated that electronegative LDL (LDL-) can induce cytokine releases and intracellular TG accumulation in a monocyte-differentiated macrophage cell line, THP-1, which is over-expressing CD14 (THP1-CD14-TLR4). In this cellular model is showed that LDL(-) induces the expression and release of pro-inflammatory cytokines as well as macrophage differentiation via TLR4, which also is blocked by HDL. This inflammatory action is similar to that promoted by oxLDL. Moreover, LDL(-) uptake produced intracellular TG accumulation in LD, which is differentiated to the effect of others modified LDLs, included aggLDL. Interestingly, this intracellular accumulation of lipids is not dependent on TLR4 and HDL. The manuscript is well structured but some considerations should be taken in order to clarify some aspects of this study.

Major comments:

1) In Fig 1(b) the symbols (*) of statistical significance are absent or the difference or not must be indicated between LDL(-) and (LDL+).

2) In the Fig 2 (a) is needed to clarify the use of the symbol # when it is indicating “vs. all LDLs”. This phrase is very understandable.

3) Lane 270: the sentence “LDL(-) induces cell differentiation” is not correctly placed in the subsection of the manuscript. This sentence should be removed.

4) In Fig 3(a) and lanes 275-276 the authors are referring that LDLs (+ and -) induces macrophage differentiation through the production of pseudopodia. It is very difficult to analyse these results without quantitative information. Thus, the number of cells producing pseudopodia must be quantified and presented together with the microscopy images in Figure 3. Moreover, in Fig 3(b) if the different level of expression of CD14 is produced by the differentiation effect of LDL(-) it should be indicated through statistical significance.

5) In Fig 4(a) is showed that anti-TLR4 antibody can block the LDL(-) effect on the macrophage differentiation, but on LDL(+) is not showed. Is it blocking or not the macrophage differentiation promoted by native LDL? It is very important that the authors refer to the effect of the anti-TLR4 in macrophages stimulated with native LDL. On the other hand, the question would be: Is it specific the effect of LDL(-) on the macrophage differentiation exerted via TLR4? In addition, here also is very necessary to quantify the pseudopodia.

6)In Figure 4(b) and 4(c) result very difficult to evaluate the inhibitory effect of anti-TLR4 and HDL on cytokine release as percentage of inhibitions (y-axis). In these cases, these inhibitions should be clearly evident showing the concentration of cytokines (similar Figure 2(a)) in the absence and presence of anti-TLR4 antibody and HDL.

7) In Fig 5(b) is showed that oxLDL has a competitive effect on the LDL(-) uptake. The question is: what occur with the competition of oxLDL with LDL(+) uptake? This control would be important to evaluate the specificity of the binding and uptake of LDL(-) in macrophages. Then, this control would be important to confirm the sentence indicated in the manuscript, lane 361-362: “LDL(-) uptake was displaced by increasing concentrations of oxLDL (Figure 5 (b)), suggesting a role for scavenger receptors in its uptake”. Moreover, the author did not mention about the native LDL uptake via LDL-R, which is important to discuss about the different pattern of lipid accumulation showed in Figures 7 and 8. Although the authors to do mention in the Discussion section about CD36 and LRP1 as possible receptors of LDL(-), would be very important an additional information about possible mechanisms of binding and uptake of LDL(-) in macrophages in order to produce different pattern of lipid accumulations compared with modified LDLs and native LDL.

8) In Figure 6 the size of cells in the panel of LDL(-) treatment must be normalized to the other panels. In the panel of macrophages of LDL(-) for 48 h is more bigger than in the other panels. In these figures the cellular size must be indicated with a reference bar (µm).

9) In lanes 396 to 397 is mentioned: “In the current study, we found that LDL (-) and aggLDL (after 60 s of vortexing) showed a 1.5-fold and 5-fold aggregation level, respectively, compared to LDL(+); ….”. This sentence is very difficult to understand. It is not clear what is comparing and where these data are shown in the manuscript or in suppl. material.

10) The comparative analysis of aggregation of LDLs by different times of vortexing is very difficult to understand without quantitative information of LDL uptake. Values of LDs formation should be included. In addition, if these data of lipid accumulation is relevant, the authors should be consider to include the Fig S2 as figure of the principal manuscript.

11) Lane 464: The authors can not do mention of “….reaching similar values” because the data were not quatified.

Minor considerations:

1) In discussion section, lanes 571-573: The last sentence is not conclusive and it should be corrected or completed.

2) Lane 583: after references [41, 42] a point should be included as final phrase and not coma.

Round 2

Reviewer 3 Report

After I read the authors' responses to my critics in the first round and other Reviewers' comments, I think I should give the manuscript a second thought. Although I am still not very convinced about the novelty issue, I think the readers may find the story interesting.

My major question here is what is the potential receptor for the LDL (-) to mediate all the effects in the manuscript? The authors provided several potential candidates (e.g. LOX1, CD36, etc) in their Discussion. It would be of high interest to further explore it in future studies. BTW, the figure they provided in the response letter should be included in the supplemental materials.

Author Response

Thank you for the opportunity to resubmit our manuscript.

We agree with the reviewer that the unresolved question in this study is the determination of the receptor mediating all the effects of LDL(-). It seems quite clear that TLR4 is involved in some of the LDL(-)-induced effects. Our hypothesis is that the activation of this receptor by LDL(-) leads to cytokine release and macrophage differentiation. However, TLR4 does not seem to mediate internalization of LDL(-) or intracellular lipid accumulation. Based on the current data, we cannot determine which are the mechanisms related to binding/uptake of LDL(-) leading to neutral lipid accumulation in macrophages. We speculate in Discussion that some receptors, such as SRA, CD36, LOX-1 and LRP1 could be involved. Once internalized, LDL(-) could activate specific pathways due to its increased content in active biomolecules.

Finding the receptor/s and intracellular pathways involved in LDL(-)-induced intracellular lipid accumulation in macrophages is our next aim. This point is of high interest to ascertain and it will provide a better knowledge of the role of LDL(-) in atherosclerosis. It has been reported that LOX-1 (Lu et al. 2009, reference 11) and LRP1 (Revuelta-Lopez et al. 2015, reference 37) are able to mediate LDL(-) effects in other cell types (endothelial cells and cardiomyocytes, respectively). In contrast, we previously reported that SRA from macrophages binds poorly LDL(-) (Benitez et al. 2004, reference 26). Regarding CD36, we have started experiments to block this receptor by using an antibody anti-CD36. However, much more detailed studies, using different blocking molecules and silencing these genes, are necessary to define the role of each receptor on LDL(-) internalization and subsequent intracellular lipid accumulation.  

According to the reviewer’s suggestion, we have added flow cytometry analysis (shown in our previous response) as Supplementary Figure 1.

Reviewer 4 Report

Minor comment:

In the Figure 1, letters (a) and (b) were omitted.

Author Response

Thank you for your observation. We have included (a) and (b) in Figure 1.